# A Facile Preparation of Sandwich-Structured Pd/Polypyrrole-Graphene/Pd Catalysts for Formic Acid Electro-Oxidation

**DOI:** 10.3390/molecules28145296

**Published:** 2023-07-09

**Authors:** Zhenjiang Lu, Wenjin Qin, Juan Ma, Yali Cao, Shujuan Bao

**Affiliations:** 1State Key Laboratory of Chemistry and Utilization of Carbon Based Energy Resources, College of Chemistry, Xinjiang University, Urumqi 830046, China; lzj318719@163.com (Z.L.);; 2Department of Science and Technology, Xinjiang University, Urumqi 830046, China; 3Institute of Clean Energy & Advanced Materials, Southwest University, Chongqing 400715, China

**Keywords:** electrodeposition, polypyrrole, Pd nanoparticles, DFAFCs

## Abstract

Direct formic acid fuel cells (DFAFCs) are one of the most promising power sources due to its high conversion efficiency; relatively low carbon emissions, toxicity, and flammability; convenience; and low-cost storage and transportation. However, the key challenge to large-scale commercial applications is its poor power performance and the catalyst’s high preparation cost. In this study, a new sandwich-structured Pd/polypyrrole-graphene/Pd (Pd/PPy-Gns/Pd)-modified glassy carbon electrode (GCE) was prepared using a simple constant potential (CP) electrodeposition technique. On the basis of the unique synthetic procedure and structural advantages, the Pd/PPy-Gns/Pd shows a fast charge/mass transport rate, high electrocatalytic activity, and great stability for formic acid electro-oxidation (FAO). The mass activity of Pd/PPy-Gns/Pd electrode reaches 917 mA·mg^−1^_Pd_. The excellent catalytic activity is mainly due to the uniform embedding of Pd nanoparticles on the polypyrrole-graphene (PPy-Gns) support, which exposes more active sites, and prevents the shedding and inactivation of Pd nanoparticles. At the same time, the introduction of graphene (Gns) in the PPy further improved the conductivity of the catalyst and accelerated the transfer of electrons.

## 1. Introduction

With the extensive use of fossil fuel energy in the world, serious environmental pollution and energy shortages have been caused, which has accelerated the development of clean energy technology. As a new type of clean energy, direct methanol fuel cells (DMFCs) have been regarded as the most promising energy conversion devices for electric vehicle applications and portable power, benefiting from high efficiency, high power density, high theoretical open circuit potential, low pollutant emission, easy transport, and safe storage of methanol as a liquid fuel [1,2,3,4]. However, its wide application is still restricted by its poor durability, high cost, operational flexibility, and reliability. Recently, direct formic acid fuel cells (DFAFCs) as a novel energy and conversion device have attracted much attention due to the following advantages: (1) Nontoxic formic acid (FA) as fuel is safer for humans and the environment than that of methanol [3,5]; (2) because of the electrostatic repulsion between -SO_3_ and HCOO-groups, the cross permeation of FA on Nafion membranes or proton exchange membranes is two orders of magnitude lower than that of methanol [6,7]; (3) FA has faster oxidation kinetics than methanol [8,9]; (4) FA is also a clean and renewable energy that can be obtained through biomass conversion processes. Additionally, recent progress in the electrochemical conversion of carbon dioxide into formic acid has directed considerable attention to DFAFCs. Therefore, as a promising energy conversion device, DFAFCs have attracted wide attention in the scientific community. However, a major challenge for DFAFCs is sluggish formic acid oxidation (FAO) and catalyst poisoning caused by the reaction intermediate species (i.e., CO) [5,10].

Compared to platinum-based catalysts, during the oxidation process of palladium-based catalysts, HCOOH is first adsorbed on its surface to promote HCOOH dissociation, and then the adsorbed intermediates are converted to CO_2_ through the 2 electrons (2e^−^) oxidation step. Because of this direct 2e^−^ oxidation, it is easy to poison the Pt-based catalyst with unwanted oxidation products such as CO, which do not occur on the Pd-based catalyst. Additionally, Pd is abundant in nature and cheap. Therefore, it is considered the most promising candidate material for DFAFC applications. However, the low activity and poor stability of palladium-based electrocatalysts hinder further commercial applications. In recent years, how to improve the intrinsic catalytic activity and stability of Pd-based catalysts has been an important challenge for researchers. It has been reported that the morphology and size of palladium nanoparticles have a great influence on their electrocatalytic activity and stability, and their activity can be adjusted through different preparation methods and loading on different carrier surfaces to make it disperse well [3,11,12].

Up until now, electrodeposition technology has been widely used on account of its simple preparation method, good repeatability, and its ability to effectively regulate the morphology composition and activity of catalysts by adjusting electrolyte composition, temperature, deposition time, current/potential, and suitable catalyst support. In particular, embedding noble metal nanoparticles on different supporting surfaces using electrodeposition technology can not only expose more active sites and improve the activity, but also enhance the stability of the catalyst [13,14]. Carbon-based materials have shown excellent application prospects for electrocatalysts. It can not only reduce the amount and costs of precious metals, but also enhance the stability of catalysts. Common carbon carriers of dispersed metal nanoparticles include activated carbon, carbon nanotubes, carbon nanofibers, graphene, etc., which all have good electrical conductivity and great stability for electrochemistry [15,16,17]. In particular, the emergence of graphene opens up a new avenue for utilizing two-dimensional (2D) carbon support, which is widely used in electrocatalyst support because of its large specific surface area (2630 m^2^ g^−1^), good chemical stability, and excellent electronic conductivity (Intrinsic electron mobility is about 200,000 cm^2^ v^−1^ s^−1^). By virtue of these attractive properties, graphene has inspired research interest in energy storage and conversion, catalysis, and electrochemical sensors [18,19,20]. For example, Feng et al. reported that a PdNi/N-doped graphene aerogel showed excellent FAO [21]. Zhang et al. reported that ultrafine Cu_1_Au_1_@Cu_1_Pd_3_ nanodots synergized with graphene nanosheets exhibited exceptional electrocatalytic activity and noble-metal utilization toward oxygen reduction, methanol oxidation, and ethanol oxidation reactions, which is mainly due to the large specific surface area and good electrical conductivity of graphene, exposing more active sites and accelerating the electron migration speed [22]. However, the chemically inert surface of graphene makes it difficult to load metal nanoparticles effectively, which usually requires surface oxidation or physical adsorption. The preparation process is very complicated, and the effect is not ideal. The conducting polymer with high electrical conductivity and excellent electrochemical stability has been used to enhance the dispersion of metal nanoparticles and the activity of the catalyst [23,24]. Therefore, by embedding graphene nanosheets into conductive polymers, the synergistic effect of the two can not only enhance the adhesion of the graphene surface and facilitate the uniform dispersion of metal nanoparticles, but also improve the conductivity of catalysts.

Based on the above discussion, we synthesized a sandwich-structured Pd/PPy-Gns/Pd using a relatively simple electrodeposition method. Firstly, the first layer of Pd nanoparticles were deposited on the glassy carbon electrode (GCE) using a constant potential (CP) technique to enhance the conductivity of the catalyst. Secondly, a small amount of graphene was dispersed in the pyrrole precursor, and then a graphene-doped polypyrrole (PPy-Gns) carrier was prepared through electrodeposition to ensure the uniform dispersion and anchor of Pd nanoparticles on its surface. Finally, another layer of Pd nanoparticles were deposited on the surface of the PPy-Gns support to expose more active sites, preventing the shedding and deactivation of the Pd catalyst, ensuring cyclic stability. Additionally, catalyst activity and stability were further regulated by adjusting the reaction time. High power transmission electron microscopy (TEM) images shows that the morphology, particle size, and dispersion of Pd particles was efficiently improved in the presence of PPy-Gns layers, and while the electrodeposition time was prolonged to 400 s, the obtained sandwich-structured Pd/PPy-Gns/Pd-modified electrode exhibited high current density and good stability, which could be mainly attributed to the metal palladium nanoparticles being uniformly dispersed on the surface of PPy-Gns, exposing more active sites. Meanwhile, its unique sandwich structure provided a good channel for the adsorption of formic acid molecules and sped up the reaction kinetics. This work offers a low-Pd-loading, highly active, and stable catalyst for direct formic acid fuel cells.

## 2. Results and Discussion

### 2.1. Morphology and Structure Characterizations

The overall synthetic route of sandwich-structured Pd/PPy-Gns/Pd-modified GCE is schematically illustrated in Figure 1. Firstly, the first layer of Pd nanoparticles was deposited on the surface of the GCE using the constant potential (CP) technique. Secondly, a small amount of graphene was dispersed in the pyrrole precursor, and then a graphene-doped polypyrrole (PPy-Gns) carrier was deposited to ensure the uniform dispersion of Pd nanoparticles. Finally, the Pd nanoparticles were uniformly deposited on the surface of the PPy-Gns carrier to expose more active sites, preventing the shedding and deactivation of Pd nanoparticles, and ensuring the cyclic stability of the catalyst. For comparison, Pd/PPy-Gns/Pd and Pd/PPy/Pd electrodes with different deposition times were prepared through similar methods, and Pd/PPy-Gns electrode deposited only the second layer of the PPy-Gns carrier and the third layer of metallic Pd nanoparticles.

To investigate the structure and composition of GO, Gns, and different Pd/PPy-Gns/Pd electrodes, it was found through SEM and TEM images that the morphology of the GO was a crumpled gauze, but compared with GO, Gns was thinner and more dispersed (Figure 2a–c), which may be because the surface of GO has oxygen-containing groups that are easy to agglomerate, proving that we have successfully synthesized ultra-thin graphene nanosheets. For the Pd/PPy-Gns/Pd composite material of the sandwich, when the electrodeposition time is 400 s, the first layer of metal Pd grows on the surface of GCE electrode (labeled as Pd/GCE), and the size of Pd nanoparticles is 100~200 nm (Figure 2d). After being further coated by PPy-Gns (labeled as PPy-Gns/Pd/GCE), the surface of the PPy-Gns/Pd/GCE became rough, and an abundant of graphene nanosheets could be observed (Figure 2e), which proved that the graphene was successfully incorporated in the PPy. Subsequently, another layer of Pd nanoparticles was deposited on the surface of the PPy-Gns/Pd/GCE (labeled as Pd/PPy-Gns/Pd/GCE), and the obtained Pd nanoparticles were evenly distributed and significantly smaller than Pd/GCE (Figure 2f). Moreover, with the extension of electrodeposition time, the size of Pd nanoparticles increased, and obvious agglomeration occurs (Figure 2g), which confirmed that the PPy-Gns is a good supporting layer here, and the metal palladium nanoparticles can be evenly dispersed by controlling the electrodeposition time. To further investigate the structure of the Pd/PPy-Gns/Pd catalyst, TEM and high-resolution TEM (HRTEM) images were observed. As depicted in Figure 2h, a large number of Pd particles grew uniformly on the surface of the PPy-Gns carrier. Meanwhile, the lattice fringes in the HRTEM diagram (in Figure 2i) correspond to the (111) crystal faces of Pd nanoparticles, which further proved that sandwich-structured Pd/PPy-Gns/Pd-modified GCE was successfully synthesized.

XRD patterns were conducted to characterize the crystal surface structure of materials. At the same time, it is very effective to determine the oxidation degree of carbon materials and the number of oxygen-containing groups by calculating the layer spacing of carbon materials through Scherrer’s formula. That is shown in Figure 3a. The pristine GO shows a peak around 9.82°, and the corresponding layer spacing is 9.4 Å. The larger layer spacing indicates that the oxidation of GO is more thorough, and that more -COOH as well as -OH is on the surface. When the GO was reduced, the diffraction peak shifted to 25.8°, corresponding to the (002) crystal plane of Gns (PDF#75-1621), and the calculated layer spacing is about 3.5 Å, indicating that GO was successfully reduced by NaBH_4_ at 90 °C for 3 h. At the same time, the oxygen-containing groups are removed, and the material returns to the ordered graphite crystal plane structure [25,26]. Finally, the sandwich-structured Pd/PPy-Gns/Pd-modified GCE increases the diffraction peak of Pd (PDF#87-0638), corresponding to diffraction peaks at 40.2 (111), 48.8 (200), and 68.3° (220). Raman spectroscopy is the most direct and effective method for characterizing the structure and properties of carbon materials. Figure 3b shows the Raman spectra of the first layer of Pd nanoparticles, the second layer of PPy-Gns, and the composite layer of Pd/PPy-Gns/Pd. For the first layer, we only observed a weak peak at 1362 cm^−1^. In the second layer of PPy and Gns, the strong band at about 1581 cm^−1^ represents the C=C backbone stretching of PPy, and the double peaks located at approximately 1040 and 1081 cm^−1^ are assigned to a C-H in-plane deformation. Another bimodal near 1350 cm^−1^ belongs to the cyclic stretching pattern of PPy, and the bands at 930 cm^−1^ and 1240 cm^−1^ are attributed to the C-H out-of-plane bending of oxidized PPy and C-H in-plane bending, respectively, which matched those previously reported for typical PPy [27,28,29]. For the graphene, a prominent G band at 1587 cm^−1^ and a typical D band at 1326 cm^−1^ were observed, which are related to defected or disordered carbon and graphitic carbon, respectively [30,31]. For the compound of Pd/PPy-Gns/Pd, both D and G bands shift to 1320 and 1580 cm^−1^, which reveals a π-π interaction between PPy and Gns sheets. Additionally, a broad peak at 1350 cm^−1^ and three other peaks at 936, 1024, and 1078 cm^−1^ are the characteristic peaks of PPy. The above results further prove that a Pd/PPy-Gns/Pd sandwich catalyst has been successfully synthesized, which is consistent with XRD results.

### 2.2. Electrocatalysis of Pd-Modified Electrode toward Formic Acid Oxidation

The inset of Figure 4a shows the typical cyclic voltammograms curves (CVs) of Pd-coated GCE, and during the anode scanning process, there are two distinct peaks in the negative potential region. The first peak at −0.18 V corresponds to the oxidation of the adsorbed hydrogen (H_ad_), while the subsequent peak at −0.08 V is due to the oxidation of the absorbed hydrogen (H_ab_) on the Pd surface, which is consistent with the reported literature [32,33]. The cathodic peak during the reverse scan is the absorption and desorption of hydrogen, and the peak at 0.45 V in the higher potential region is the reduction peak of produced Pd oxide. It is well known that the unique voltammetric features in the hydrogen region reflect the better catalytic performance of Pd catalysts. Another curve in Figure 4a is the CV of Pd-coated GCE in 0.5 M H_2_SO_4_ containing 0.25 M HCOOH solution, and an obvious formic acid oxidation peak at 0.14 V is displayed in the presence of HCOOH, which indicates that the electrodeposition of Pd on GCE has excellent catalytic activity for formic acid oxidation. Since catalyst support plays a key role in catalyst activity and durability, in order to further improve the utilization of palladium and the electrocatalytic activity of formic acid, polypyridine and graphene were introduced, and different modified electrodes were designed and characterized as follows:

The CVs of different electrodes in 0.5 M H_2_SO_4_ + 0.25 M HCOOH aqueous solutions are displayed in Figure 4b. The oxidation peak of formic acid on Pd-coated GCE (Pd/coated) is displayed clearly (304 mA mg^−1^_Pd_), while when coated by PPy-Gns (Pd/PPy-Gns), the oxidation peak of formic acid disappeared, implying that the PPy-Gns layer on the Pd surface blocks Pd catalytic active sites. Interestingly, as another layer of Pd is anchored on the surface of the PPy-Gns/Pd-modified electrode (Pd/PPy-Gns/Pd), the oxidation peak current increases once again, and the electrocatalytic activity of Pd/PPy-Gns/Pd is about three times higher than that of Pd-coated GCE. This result strongly demonstrated that the sandwich-structured Pd layer interfaced with the conductive polymer-graphene layer is a feasible way to design catalysis with high performance. To investigate the electrocatalytic activity of Pd on various substrates, differently structured electrodes were prepared and measured. Figure 4c shows that the oxidation peak current of the PPy-Gns/Pd composite electrode (873 mA mg^−1^_Pd_) is slightly decreased than that of the Pd/PPy-Gns/Pd (917 mA mg^−1^_Pd_). This may be due to the first Pd layer, which has excellent electronic conductivity and an accelerated charge transfer. However, when graphene is not doped in pyrrole, the oxidation peak current of FAO decreases significantly. The results strongly proved the PPy-Gns composite is a good support with excellent electronic conductivity, presenting uniform particle size distribution and a high surface area for Pd particles.

Because the electrodeposition time affects the size and dispersion of Pd nanoparticles, in order to explore the influence of electrodeposition time on the catalysts, the Pd/PPy-Gns/Pd-modified electrodes were obtained at different times. While the deposition potential was kept at −0.4 V, the oxidation peak current density increased gradually with the deposition time. However, when the deposition time reached 600 s, the oxidation peak currents decrease. This may be because when the deposition time increases, the amount and particle size of Pd also increases accordingly, but the increasing of particles size reduces the real electrochemically active area of Pd. Hence, when the deposition time reached 400 s, the catalyst exhibited the highest catalytic activity, and the mass activity reaches 917 mA mg^−1^_Pd_. (in Figure 4d). Electrochemical impedance spectroscopy (EIS) is an important parameter for determining the performance of electrocatalysis. The measurements were carried out to compare the characteristics of ion and charge transfer in differently structured electrodes. In the Nyquist plots, the semicircular portion of the impedance spectrum at a higher frequency represents the charge transfer-limited process, and the diameter of the semicircle equals to the surface charge transfer resistance (R_ct_), which usually represents the resistance of electrochemical reactions on the electrode, and is called the Faraday resistance; the straight line is ascribed to the diffusive resistance of the electrolyte in electrode pores and the proton diffusion in host materials [34,35]. The simulated equivalent circuit diagram is illustrated in Figure 4e. R_s_, R_ct_ (R_1_, R_2_), and CPE (CPE_1_, CPE_2_) were assigned to solution resistance, charge transfer resistance, and capacitive reactance, respectively. The R_ct_ value shows that the semicircle of the Pd/PPy-Gns/Pd has a smaller charge transfer resistance (R_ct_ = 11.3 Ω) than PPy-Gns/Pd (R_ct_ = 15.9 Ω) and Pd/PPy/Pd (R_ct_ = 32.1 Ω). This indicates that Gns-doped PPy can improve electrical conductivity, accelerating catalytic reaction kinetics. These results illustrate that PPy-Gns not only serve as a supporting material for Pd deposition, but that it can also increase the electron transfer rate. The stability of a catalytic electrode is another important parameter to consider for commercial applications. Figure 4f shows the chronoamperometry curves (i-t) of the electrodes with different structures. After the reaction lasted for 7200 s, the polarization current of FAO on each catalyst showed a rapid decline in the initial stage due to the adsorption of CO. Subsequently, the current densities decrease slowly and reach a pseudo-steady state. Finally, the Pd coated GCE exhibits a smaller current, and the Pd/PPy-Gns/Pd-coated GCE shows the highest current density, which proved that the Pd/PPy-Gns/Pd-coated GCE has good catalytic stability for FAO, consistent with the test results of the CV curve.

To further evaluate the intrinsic activity of FAO fuel cells, the electrochemically active surface area (ECSAs) is usually evaluated to explore the influence of the structural composition on the intrinsic activity of the catalyst, and the electrochemical double-layer capacitances (C_dl_) obtained by CVs at different sweep speeds is positively correlated with the ECSAs. Therefore, C_dl_ can be used to evaluate the number of electrochemically active sites exposed to different materials [36,37]. As can be seen from Figure 5, Pd/PPy-Gns/Pd-modified GCE shows the largest double layer capacitance (26.3 mF cm^−1^), which proves that it has a large electrochemically active area, and has good formic acid oxidation activity, which is consistent with the test results of CV curve.

## 3. Materials and Methods

### 3.1. Chemicals

Palladium chloride (PaCl_2_, 98%), pyrrole (C_4_H_5_N, 98%), formic acid (HCOOH, 98%), and Nafion solution were purchased from Aladdin Ltd. Graphite powder, potassium permanganate (KMnO_4_, 98%), concentrated sulfuric acid (H_2_SO_4_, 98.3%), concentrated nitric acid (HNO_3_, 98%), and sodium nitrate (NaNO_3_, 99%) were purchased from Shanghai Chemical Reagent Factory. Pyrrole was purified by distillation under reduced pressure before polymerization. All other chemicals are analytical grade and could be used without further purification.

### 3.2. Preparation of Graphene Oxide and Graphene

The graphene oxide (GO) was prepared according to the modified Hummer’s method [38]. Specifically, 4 g of graphite powder and 3 g of NaNO_3_ were added into 230 mL of H_2_SO_4_ and pre-oxidized in an ice bath for 10 min. Then, 19 g of KMnO_4_ was slowly added. In order to prevent overheating and explosion, it was necessary to stir continuously for 5 days below 20 °C to ensure complete oxidation. The obtained precipitate is centrifugally washed with distilled water to a pH of 6–7. Finally, the samples were freeze-dried at −45 °C for 48 h and labeled with graphene oxide (GO). The graphene (Gns) was obtained through reduction of GO with NaBH_4_ according to the literature [39]. Specifically, 100 mL of distilled water was accurately measured and ultrasonically dispersed with 10 mg of graphene oxide for 1 h. Then, 200 mg of NaBH_4_ was added for magnetic stirring for 30 min, then heated to 90 °C for 3 h so that the brown-yellow flocculant slowly turned into a black precipitate. Finally, it was centrifuged with distilled water and vacuum dried at 55 °C for 12 h. The final product was labeled Gns.

### 3.3. Preparation of Pyrrole-Graphene Mixtures

24 mg of graphene was dispersed in 5 mL of pyrrole and refluxed at 140 °C for 5 h under N_2_ protection to obtain a pyrrole-graphene mixed solution containing 0.5 wt% of graphene.

### 3.4. Preparation of Modified Electrodes

The GCE with a diameter of 3 mm was polished with 0.3 μm and 0.05 μm of alumina slurry, and then subjected to ultrasonic treatment in deionized water and ethanol several times to obtain a smooth mirror surface before each experiment. An electrodeposition technique was used to prepare modified electrodes. The first layer and third layer of Pd nano-particles on modified electrodes were prepared in 0.5 M HCl solution containing 0.1 M PdCl_2_ using a constant potential (CP) technique at −0.4 V. The PPy and PPy-Gns layer on modified electrodes were prepared in 0.5 M H_2_SO_4_ containing 0.2 M pyrrole-graphene solution using a CP technique at 0.7 V for 100 s. The obtained electrodes were rinsed with deionized water before further measurements.

### 3.5. Chemoelectrochemical Performance Test of Formic Acid Oxygen

All electrochemical experiments were carried out at a CHI 760D electrochemical workstation (Shanghai Chenhua, China), using a traditional three-electrode cell composed of modified GCEs as the working electrode, a platinum electrode as the auxiliary electrode, and a saturated calomel electrode (SCE) as the reference. In general, 2.5 mg of catalyst was dispersed in a mixture of 150 μL deionized water, 340 μL ethyl alcohol, and added to 10 μL (5 wt%) of Nafion. Subsequently, the catalyst suspension of 10 μL was uniformly dropped on the surface of GCEs and dried at room temperature for 60 min. The catalytic activity of different electrodes for FAO was studied by a CV and carried out in a solution of 0.5 M H_2_SO_4_ containing 0.25 M HCOOH at a scan rate of 50 mV s^−1^. Before each test, the solution was saturated with nitrogen. 

### 3.6. Characterization

The microstructure and morphology of different modified electrodes were investigated using scanning electron microscopy (SEM, JSM-6300, JEOL, Tokyo, Japan), transmission electron microscopy (TEM, JEM-2100F JEOL, Tokyo, Japan), x-ray diffraction (XRD, Rigaku, SmartLab, Tokyo, Japan), and Raman spectroscopy (Bruker Senterra R200-L, Billerica, MA, USA).

## 4. Conclusions

In summary, we have constructed a sandwich-structured Pd/PPy-Gns/Pd-modified GCE using a simple CP electrodeposition technique. Because of the unique structural advantages, the three-dimensional Pd/PPy-Gns/Pd catalyst shows fast charge/mass transport rate, high electrocatalytic activity, and great stability for FAO. The excellent catalytic activity is mainly due to the uniform embedding of Pd nanoparticles on the PPy-Gns support, which exposes more active sites, and prevents the shedding and inactivation of Pd nanoparticles. Meanwhile, the introduction of Gns in the PPy further improved the conductivity of the catalyst and accelerated the transfer of electrons. The excellent electrochemical performance of Pd/PPy-Gns/Pd combined with the relatively low price and abundance of Pd makes it a strong candidate for replacing Pt, which will promote the development of fuel cells. More importantly, it has also proposed an effective and universal (electro)catalyst-design paradigm for various energy-conversion or environmental (electro)catalytic reactions.

## Figures and Tables

**Figure 1 molecules-28-05296-f001:**
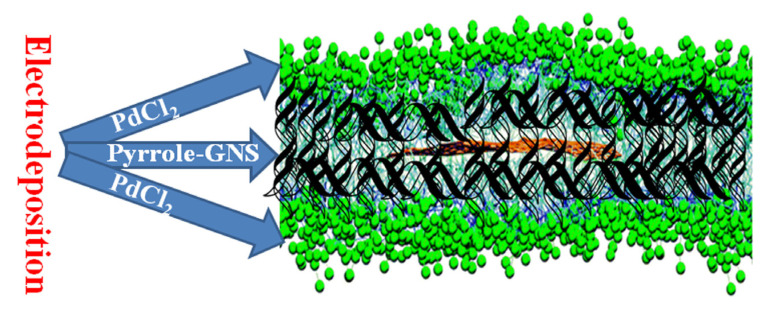
Schematic illustration of the synthesis strategy of Pd/PPy-Gns/Pd.

**Figure 2 molecules-28-05296-f002:**
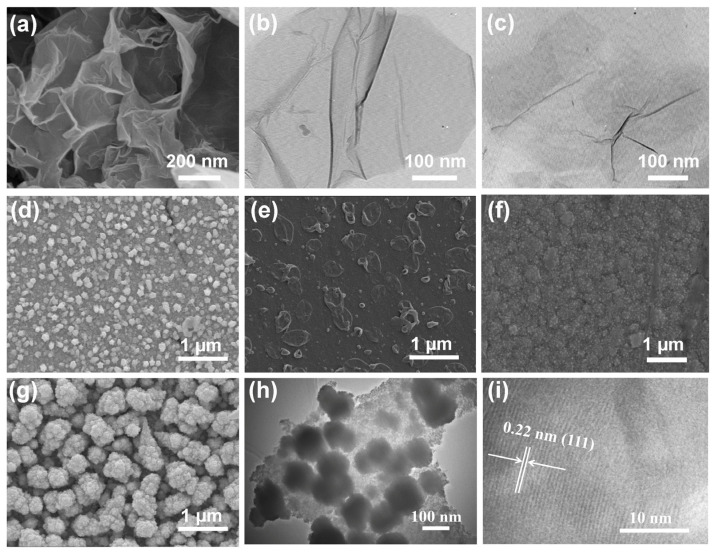
(**a**) SEM of GO, (**b**,**c**) TEM of GO and graphene respectively, (**d**–**f**) SEM of Pd, PPy-Gns/Pd and Pd/PPy-Gns/Pd-modified GCE at electrodeposition time of 400 s, (**g**) Pd/PPy-Gns/Pd-modified GCE at electrodeposition time of 600 s, (**h**) TEM and (**i**) HRTEM of Pd/PPy-Gns/Pd-modified at electrodeposition time of 400 s.

**Figure 3 molecules-28-05296-f003:**
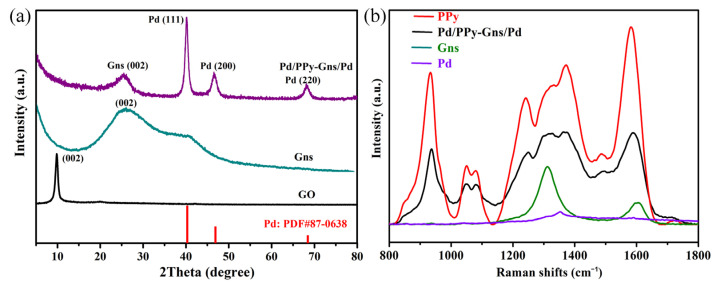
(**a**) XRD patterns, (**b**) Raman spectra of the as-prepared samples.

**Figure 4 molecules-28-05296-f004:**
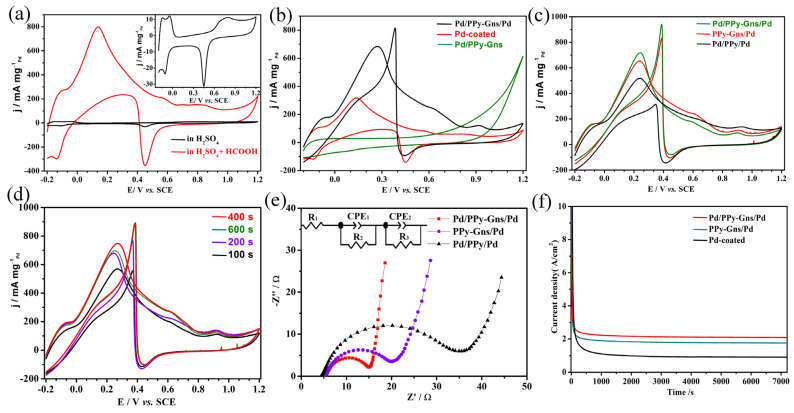
(**a**) CVs obtained on the Pd-coated GCE in 0.5 M H_2_SO_4_ and 0.5 M H_2_SO_4_ containing 0.25 M HCOOH, (**b**–**d**) CV obtained on various Pd-based catalysts-coated GCE and different deposition time in 0.5 M H_2_SO_4_ + 0.25 M HCOOH, (**e**) Nyquist plots of different electrodes at open circuit potential, (**f**) i-t curves at −0.2 V vs. SCE.

**Figure 5 molecules-28-05296-f005:**
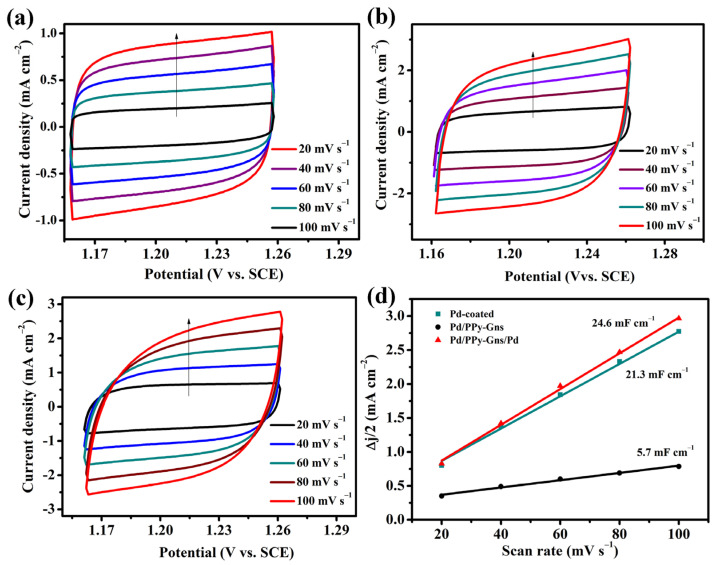
(**a**–**c**) The FAO CV cures of Pd/coated, Pd/PPy-Gns, Pd/PPy-Gns/Pd-modified GCE with different scanning speeds in 0.5 M H_2_SO_4_ + 0.25 M HCOOH, (**d**) corresponding C_dl_ value.

## Data Availability

Not applicable.

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
