# Peer review of "A Facile Preparation of Sandwich-Structured Pd/Polypyrrole-Graphene/Pd Catalysts for Formic Acid Electro-Oxidation"

_molecules, 2023, doi:10.3390/molecules28145296_

Round 1
Reviewer 1 Report
Thanks a lot for the opportunity to review the manuscript titled " A facile preparation of sandwich-structured Pd/Polypyrrole-Graphene/Pd catalysts for formic acid electro-oxidation". In this paper, the author suggested a straightforward method for creating a specific type of catalyst. The catalyst consists of a sandwich structure with Pd nanoparticles embedded between layers of polypyrrole-graphene composite. The purpose of this catalyst is to facilitate the electro-oxidation of formic acid, which is a crucial reaction in direct formic acid fuel cells (DAFCs). The subject area of the manuscript is quite interesting, and it would certainly add a scientific contribution to the relevant field. I recommend the publication in "Molecules" after minor revision. The following suggestions are provided for the authors' revising manuscript.
1. Provide context: Start the abstract by briefly explaining the significance of DAFCs as a promising energy source for the low-carbon economy. This will help readers understand the relevance and importance of the study.
2. Simplify and reorganize the sentence structure: Some sentences in the abstract are long and complex. Try to break them down into shorter sentences to improve readability. Additionally, consider reorganizing the information to present a logical flow of ideas.
3. To improve the quality of Introduction Section…can you elaborate on the advantages of direct formic acid fuel cells (DFAFCs), particularly in terms of the safety of using nontoxic formic acid as fuel compared to methanol, the reduced cross permeation on Nafion membranes, and the faster oxidation kinetics of formic acid?
4. When discussing carbon-based materials as support for electrocatalysts, such as graphene, could you provide more specific examples of their excellent application prospects and highlight their unique properties, such as large specific surface area, good chemical stability, and excellent electronic conductivity?
5. Can you provide more specific results or evidence from the study, such as the observed high current density and stability of the sandwich-structured catalyst, to further support its potential for direct formic acid fuel cells?
6. In Figure 3a, the authors should have to mention all the planes.
7. The inset in Figure 4a is not clearly visible.
8. In EIS, the authors did not mention the Rct values. I recommend to use EIS Fitting with some software.
9. Purity of using materials must be clear.
10. To add more value into the introduction and literature part, please cite latest references. The following papers should be added to REFERENCES:
DOI: 10.1016/j.cej.2021.128453, DOI: 10.1016/j.foodchem.2022.133642,
DOI: 10.1016/j.teac.2021.e00138, DOI: 10.1021/acsami.1c07067
Can you discuss the broader implications of the study's findings in Conclusion section for the development and utilization of DFAFCs, particularly in terms of their role in achieving a low-carbon economy and addressing environmental concerns associated with traditional energy sources?
Typo errors need to be corrected throughout the manuscript
Reviewer 2 Report
Line 45, what FO stands for??
section 3.3. must be re-written to improve the expression
Sections 2.1 and 2.2 must be improved and made more descriptive
Conclusions should be improved, it has grammar mistakes and the authors should give sound conclusions.
Well written. However, in the discussion part grammar mistakes makes it difficult to understand. It is suggested to improve the discussion part.
